# Effects of Sodium Nitrate and Coated Methionine on Lactation Performance, Rumen Fermentation Characteristics, Amino Acid Metabolism, and Microbial Communities in Lactating Buffaloes

**DOI:** 10.3390/microorganisms11030675

**Published:** 2023-03-07

**Authors:** Yanxia Guo, Zexiang Fan, Mengwei Li, Huade Xie, Lijuan Peng, Chengjian Yang

**Affiliations:** Key Laboratory of Buffalo Genetics, Breeding and Reproduction Technology, Ministry of Agriculture and Guangxi, Buffalo Research Institute, Chinese Academy of Agricultural Sciences, Nanning 530001, China

**Keywords:** sodium nitrate, methionine, rumen fermentation, amino acid metabolism, rumen microorganism

## Abstract

Sodium nitrate is used as a non-protein nitrogen supplement while methionine is considered as a common methionine additive for ruminants. This study investigated the effects of sodium nitrate and coated methionine supplementation on milk yield, milk composition, rumen fermentation parameters, amino acid composition, and rumen microbial communities in lactating buffaloes. Forty mid-lactation multiparous Murrah buffaloes within the initial days in milk (DIM) = 180.83 ± 56.78 d, milk yield = 7.63 ± 0.19 kg, body weight = 645 ± 25 kg were selected and randomly allocated into four groups (N = 10). All of animals received the same total mixed ratio (TMR) diet. Furthermore, the groups were divided into the control group (CON), 70 g/d sodium nitrate group (SN), 15 g/d palmitate coated L-methionine group (MET), and 70 g/d sodium nitrate +15 g/d palmitate coated L-methionine group (SN+MET). The experiment lasted for six weeks, including two weeks of adaption. The results showed that most rumen-free amino acids, total essential amino acids, and total amino acids in Group SN increased (*p* < 0.05), while the dry matter intake (DMI) and rumen acetate, propionate, valerate, and total volatile fatty acids (TVFA) in Group MET decreased (*p* < 0.05). However, there was no significant difference in milk yield, milk protein, milk fat, lactose, total solid content, and sodium nitrate residue in milk among groups (*p* > 0.05). Group SN+MET had a decreased rumen propionate and valerate (*p* < 0.05), while increasing the Ace, Chao, and Simpson indices of alpha diversity of rumen bacteria. Proteobacteria and Actinobacteriota were significantly increased (*p* < 0.05) in Group SN+MET, but Bacteroidota, and Spirochaetota were decreased (*p* < 0.05). In addition, Group SN+MET also increased the relative abundance of *Acinetobacter*, *Lactococcus*, *Microbacterium*, *Chryseobacterium*, and *Klebsiella*, which were positively correlated with cysteine and negatively correlated with rumen acetate, propionate, valerate, and TVFA. Rikenellaceae_RC9_gut_group was identified as a biomarker in Group SN. Norank_f__UCG-011 was identified as a biomarker in Group MET. *Acinetobacter*, *Kurthia*, *Bacillus*, and *Corynebacterium* were identified as biomarkers in Group SN+MET. In conclusion, sodium nitrate increased rumen free amino acids, while methionine decreased dry matter intake (DMI) and rumen volatile fatty acids. The combined use of sodium nitrate and methionine enriched the species abundance of microorganisms in the rumen and affected the composition of microorganisms in the rumen. However, sodium nitrate, methionine, and their combination had no significant effect on the milk yield and milk composition. It was suggested that the combined use of sodium nitrate and methionine in buffalo production was more beneficial.

## 1. Introduction

Rumen-free amino acids and volatile fatty acids mainly come from the degradation of dietary protein and dietary fiber [1]. The degradation of microbial protein is one of the sources of free amino acids in rumen [2], but the synthesis of microbial protein is low and stable [3]. Rumen microorganisms are more inclined to decompose and utilize dietary protein into free amino acids and ammonia nitrogen. Sodium nitrate is a common methane inhibitor and non-protein nitrogen supplement [4,5]. Studies have found that sodium nitrate increases rumen redox potential [6], and changes rumen microbial structure [7] and the fermentation mode of some nutrients in the rumen [8]. Recent studies have found that sodium nitrate inhibits the biological hydrogenation of unsaturated fatty acids by rumen microorganisms [9].

Furthermore, palmitate-coated methionine is a common methionine additive for ruminants. Its degradation rate in the rumen is about 30% [10]. Methionine is an important microbial growth factor [11]. Studies have found that methionine has a positive effect on the growth of bacteria, but the mechanism is not clear [12]. In vitro studies have shown that methionine can increase ruminal asparagine content and *Compylobactor* population [13]. In addition, methionine, as the first limiting amino acid for dairy cows in the lactation period, can improve the utilization efficiency of nitrogen for dairy cows, and has synergy with dietary non-protein nitrogen supplements [14].

The unadapted buffalo rumen ingests a large amount of sodium nitrate at one time, which is easy to cause the accumulation of sodium nitrite in the rumen, causing hypoxia in animal tissues, and even death [15]. In addition, sodium nitrite can decompose nitric oxide, hydroxylamine, hydrazine, imine oxide, etc. [16]. Sodium nitrite and nitric oxide have toxic side effects on liver (*ctenopharyngodon idellus*) [17]. However, adenosylmethionine is the active form of methionine in vivo, which is of great significance for enhancing the detoxification ability of liver [18]. We hypothesized that sodium nitrate and methionine may have strong complementarity in ruminant health, which can promote its production performance. In this study, urea was used to balance the nitrogen content in the diet of buffaloes in each group, to eliminate the influence of non-protein nitrogen in the diet, and to study the effect of the combined use of sodium nitrate and methionine on the milk yield, milk composition, rumen fermentation parameters, and rumen bacteria of buffaloes, to provide a scientific basis for the combined use of sodium nitrate and methionine in the diet of buffaloes.

## 2. Materials and Methods

### 2.1. Diet and Animal Management

A total of 40 mid-lactation multiparous Murrah buffaloes within the initial days in milk (DIM) = 180.83 ± 56.7 d, milk yield = 7.63 ± 0.19 kg, body weight = 640 ± 25 kg were randomly allocated into four groups (n = 10). The control group (Group CON) was fed on a total mixed ratio (TMR), while the other three treatment groups were fed on TMR supplemented with 70 g/d sodium nitrate (>99% purity; 16.47% nitrogen content; Baishi Chemical Reagent Co., Ltd., Tianjin, China) (Group SN), 15 g/d palmitate coated L-methionine (50% purity; Sigma-Aldrich, St. Louis, MS, USA) (Group MET), and sodium nitrate group 70 g/d + palmitate coated L-methionine 15 g/d (Group SN+MET). The four different TMRs were adjusted as isonitrogenous and isoenergetic diets by adding urea (>99% purity; 46.4% nitrogen content; Baishi Chemical Reagent Co., Ltd., Tianjin, China). The nutritional level of the experimental diet meets NRC (2001), and its composition and nutritional level are shown in Table 1. Feeding and milking were carried out two times per day (07:00 and 14:00). The experiment lasted 6 weeks, including the first 2 weeks of adaptation and 4 weeks of the formal period.

### 2.2. Determination of Dry Matter Intake (DMI), Milk Yield, and Milk Composition

The dry matter intake (DMI) was measured every day during the formal period. The DMI was calculated daily by the TMR and refusals every day. The daily milk yield of each buffalo was recorded twice a day during the adaptation period and the formal period, measured at 8:00 and 16:00 every day, and the 3.5% fat-corrected milk yield (3.5%FCM) was calculated. Milk samples for determination of milk composition were collected weekly for four consecutive weeks during the formal period; 50 mL fresh milk samples were used to analyze milk composition for morning and evening separately. The remaining milk samples were mixed evenly in the same volume in the morning and afternoon as analytical samples for the determination of sodium nitrate residue content. The milk composition was determined through the milk composition tester (Foss Electric, Hildesforth Electric Company, Hillerød, Denmark), including protein, fat, lactose, and total solid content. The determination of sodium nitrate residues in buffalo milk was conducted by spectrophotometry provided in GB5009.33-2010 [19]. The calculation formula of lactation efficiency is as follows: Lactation efficiency (%) = Milk yield (kg/d) ÷ DMI (kg/d) × 100%. The 3.5% fat-corrected milk yield (3.5% FCM) was calculated by using the equation proposed given by Parbkh [20].
3.5% FCM = 0.35M + 18.57F(1)
where, 3.5% FCM is fat-corrected milk production at 3.5% fat (kg/d), M is the weight of milk (kg/d), and F is the fat content in milk (kg/d).

### 2.3. Determination of Rumen Fermentation Parameters

On the evening of the 41st day of the experimental period, buffaloes were forbidden to feed for 12 h. Rumen content samples (500 mL, both solid and fluid phase) were collected only once, at the last day of the experiment before the morning feeding, using a stomach tube. After collection, the samples were directly transported to the laboratory. About 8 mL of rumen fluid was used for the determination of microbial protein (MCP) content through colorimetry by using an ultraviolet-visible spectrophotometer (PE lambda 35, Shanghai Pudi Biotechnology Co., Ltd., Shanghai, China). For the determination of ammonia nitrogen (NH_3_-N), 4 mL of the rumen fluid was mixed with 4 mL of 0.2 mol HCl acidified and stored at −20 °C until further analysis. Later on, NH_3_-N content was measured using the indophenols method through a UV-Vis spectrophotometer (PE lambda 35, Shanghai Pudi Biotechnology Co., Ltd. Shanghai, China) at 560 nm wavelength [21]. The VFA content was determined by mixing 1 mL rumen fluid and 0.5 mL metaphosphoric acid (8.2%) and then centrifuging at 20,000× *g* (4 °C) for 10 min. After centrifugation, 920 µL of supernatant was added to 80 µL internal standard crotonic acid (1 mol/L). Different VFA fractions (C2, C3, C4, C5, iC4, and iC5) were measured using the GC system as described previously [22].

### 2.4. Determination of Amino Acid Concentration

The rumen content samples collected on the last day of the experiment were taken out. First, 5 mL of the rumen fluid was mixed and hydrolyzed with 5 mL of HCl (6 mol/L) in a constant temperature oven at 110 °C for 22 h. Then, concentrations of individual amino acids were determined through liquid chromatography-tandem mass spectrometry (LC-MS/MS) analysis using a SCIEX Triple Quad 5500 LC-MS/MS System (AB SCIEX (Pvt.) Ltd., Framingham, MA, USA) as reported previously [23]. The cation exchange column was used for amino acid analysis. The column temperature was 65 °C, and the elution gradient was 100% A-100% B linear gradient. The detector was a Waters 470 fluorescent detector. Finally, the percentage content of each amino acid was calculated according to the peak area of each amino acid in the chromatogram.

### 2.5. 16S rDNA Gene Sequencing and Bioinformatic Analysis

The rumen content samples collected on the last day of the experiment were taken out. The DNA was extracted from 1 mL of frozen rumen content (both solid and fluid phase) using the CTAB bead-beating method [24]. The quality and concentration of DNA were determined by a Nanodrop spectrophotometer (Nanodrop ND-2000, Beijing Xinxing Qiangsen Biotechnology Co., Ltd., Beijing, China). High throughput (Illumina MiSeq PE300) sequencing of the 16S rRNA gene was carried out using barcoded primers for the V3–V4 region. Based on the original data obtained by the illuminamiseqtm sequencing platform, the paired reads were spliced into a sequence according to the overlapping relationship between PE Reads. Then, the samples were identified and distinguished according to the barcode tag sequence and primer sequence at the beginning and end of the sequence to obtain each data sample. Finally, the quality of each data sample and the effect of the merge were filtered by quality control to obtain the effective sequence of each sample. The non-repetitive sequences were clustered at a 97% similarity level to obtain operational taxonomic units (OTU). Each species was compared with the OTU database using the search representative software, and then the OTU was used to classify each species. After classification, OTU abundance was obtained according to the number of sequences in each OTU. Rumen bacterial composition of samples was determined by species annotation and abundance analysis, and further alpha diversity analysis was conducted to determine the differences among samples. Binformatic analysis of the OTU data was conducted through the Meiji biological cloud platform (http://login.majorbio.com/ (accessed on 11 November 2022)) provided by Shanghai Meiji Biotechnology Co., Ltd. (Shanghai, China) to determine the relative abundance, microbial diversity matrices, and other parameters.

### 2.6. Statistical Analysis

The experimental design of the completely randomized design with only a fixed treatment effect was adopted. Data were analyzed by the analysis of variance (ANOVA) using a general linear model in SPSS software (SPSS, 2008). Statistical significance was declared at *p* < 0.05. The Alpha diversity index was calculated by Mothur software. The microbial Beta diversity was determined through principal coordinates analysis (PCoA). The linear discriminant analysis (LDA) effect size (LEfSe) was used to identify predominant bacterial taxa in each treatment group that can be considered biomarker taxa. In the present study, bacterial taxa having LDA scores (log10) > 4 were considered significantly different. Correlation analyses between ruminal fermentation and amino acid parameters with the top 28 genera of microbiota were determined using Spearman’s correlation (r). The asterisk sign was used when the r value was greater than 0.1 and the *p* values were less than 0.05 (* *p* < 0.05, ** *p* < 0.01, *** *p* < 0.001).

## 3. Results

### 3.1. Milk Yield and Composition

The dry matter intake (DMI) of buffalo in Group MET was significantly lower than that in other groups (*p* = 0.001) (Table 2). However, there was no significant difference in milk yield, 3.5% fat-corrected milk yield, lactation efficiency, milk protein, milk fat, lactose, total solid content, and sodium nitrate residue in milk among groups (*p* > 0.05).

### 3.2. Rumen Fermentation Characteristics

Treatment did not affect rumen pH, NH_3_-N, MCP, and the acetate-to-propionate ratio (A/P) (*p* > 0.05) (Table 3). However, the concentrations of acetate, propionate, valerate, and TVFA in Group MET were significantly lower than those in Groups CON and SN (*p* < 0.05). The concentrations of propionate and valerate in Group SN+MET were significantly lower than those in Groups CON and SN (*p* < 0.05). There was no significant difference in all rumen fermentation parameters in Groups MET and SN+MET (*p* > 0.05).

### 3.3. Ruminal Amino Acids

Group SN showed higher histidine, glutamine, glutamate, leucine, lysine, methionine, phenylalanine, threonine, isoleucine, tyrosine, serine, aspartic acid, essential amino acids, and total amino acids than those in other groups (*p* < 0.05) (Table 4). In addition, the concentrations of valine, proline, and non-essential amino acids were higher in Group SN as compared to Group SN+MET (*p* < 0.05).

### 3.4. Ruminal Bacterial Communities

#### 3.4.1. Rumen Bacterial Diversity

A total of 3352 OTUs were identified in the four groups (Figure 1). Among them, 1695 OTUs were found in all groups, accounting for 50.57% of the total OTUs. The highest number of OTUs (2926) was found in Group SN+MET followed by SN (2461), CON (2281), and MET (2228). The highest number of unique OTUs (585) was observed in Group SN+MET followed by SN (106), MET (57), and CON (45).

Coverage index values in this sequencing analysis were all greater than 98%, close to 1 (Table 5), indicating that the sequences of the samples had basically been detected; that is, the sequencing results can reflect the real situation of microbial species in buffalo rumen fluid samples. Group SN+MET increased (*p* = 0.001) the Ace, Chao, and Simpson indices of alpha diversity of rumen bacteria as compared to the other groups (*p* < 0.05). Based on the PCoA graph (Figure 2), the cloud of Group SN+MET was separated from the other three groups.

#### 3.4.2. Relative Abundance of Bacterial Populations

The effects of different treatments on microorganisms in the rumen contents of buffalo at the phylum level and genus level are shown in Figure 3. The dominant bacteria in the buffalo rumen were mainly Bacteroidota, Firmicutes, and Proteobacteria, which accounted for more than 86% of the whole bacteriome (Figure 3a). Other major bacterial phyla were Actinobacteriota, Cyanobacteria, Spirochaetota, and Verrucomicrobia. The relative abundance of Bacteroidota in SN+MET was significantly reduced compared with other groups (*p* = 0.004) (Figure 3a; Appendix A). However, the relative abundances of Proteobacteria and Actinobacteriota were increased in SN+MET (*p* < 0.05). Compared with the CON and SN groups, the relative abundance of Spirochaetota was increased in MET, but it was decreased in SN+MET (*p* = 0.011).

The relative abundance of major bacterial genera was shown in Figure 3b. Compared with the CON, SN, and MET groups, the relative abundances of *Prevotell*, Prevotellaceae_UCG-001, *Succiniclasticum*, norank_f__UCG-011, Prevotellaceae_UCG-003, Christensenellaceae_R-7_group, and NK4A214_group were decreased in SN+MET, but *Acinetobacter*, *Bacillus*, *Corynebacterium*, *Kurthia*, *Pedobacter*, *Lactococcus*, *Microbacterium*, *Chryseobacterium*, *Klebsiella*, *Staphylococcus*, *Enhydrobacter*, *Pseudomonas*, and *Solibacillus* were increased in SN+MET (*p* < 0.05) (Figure 3b; Appendix A). The relative abundance of Rikenellaceae_RC9_gut_group in SN significantly increased compared with other groups (*p* = 0.024). The relative abundance of *Treponema* in SN and SN+MET was decreased compared with CON and MET (*p* = 0.018). The relative abundance of *Butyrivibrio* in MET significantly increased compared with other groups (*p* = 0.031). The relative abundance of norank_f__Bacteroidales_RF16_group in MET and SN+MET was decreased compared with CON and SN (*p* = 0.009).

#### 3.4.3. Biomarker Bacteria Taxa and Metagenomic Functional Profile

We identified bacterial taxa that were predominantly abundant as biomarkers among the treatment groups through LEfSe. A total of 37 significant taxonomic clades (LDA score > 4.0) were identified with 7 genera biomarkers (Figure 4). *Prevotella* was identified as a biomarker in Group CON. Rikenellaceae_RC9_gut_group was identified as a biomarker in Group SN. Norank_f__UCG-011 was identified as a biomarker in Group MET. Four biomarker taxa including *Acinetobacter*, *Kurthia*, *Bacillus*, and *Corynebacterium* were identified as biomarkers in Group SN+MET.

#### 3.4.4. Association of Rumen Bacteria with Ruminal Fermentation Parameters and Amino Acid Contents

Our findings revealed four bacterial genera (*Klebsiella*, *Chryseobacterium*, *Lactococcus*, and *Acinetobacter*) showed a negative correlation with acetate, propionate, valerate, and TVFA (*p* < 0.05) (Figure 5). *Klebsiella* was positively correlated with the acetate-to-propionate ratio (*p* < 0.01). *Saccharofermentants* was positively correlated with acetate (*p* < 0.05). Norank_f__Prevotellaceae was positively correlated with propionate, but it was negatively correlated with the acetate-to-propionate ratio (*p* < 0.05). Norank_f__Bacteroidales_RF16_group was positively correlated with valerate (*p* < 0.05). *Pedobacter* and *Bacillus* were negatively correlated with acetate (*p* < 0.05). Prevotellaceae_UCG-003 and norank_f__F082 were negatively correlated with NH_3_-N (*p* < 0.05). Rikenellaceae_RC9_gut_group was negatively correlated with NH_3_-N and MCP (*p* < 0.05). *Acinetobacter* was positively correlated with MCP (*p* < 0.05).

Spearman’s correlation between the relative abundance of bacterial genera and ruminal amino acid contents is shown in Figure 6. Our findings revealed that 10 bacterial genera (*Staphylococcus*, *Klebsiella*, *Chryseobacterium*, *Microbacterium*, *Lactococcus*, *Pedobacter*, *Kurthia*, *Corynebacterium*, *Bacillus*, and *Acinetobacter*) showed a positive correlation with cysteine (*p* < 0.05). Prevotellaceae_UCG-003 and norank_f__norank_o__Gastranaerophilales were positively correlated with aspartic acid (*p* < 0.05). *Chryseobacterium* and *Lactococcus* showed a negative correlation with proline and phenylalanine, but Norank_f__Bacteroidales_RF16_group showed a positive correlation with them (*p* < 0.05). Veillonellaceae_UCG-001 showed a positive correlation with glycine and glutamate (*p* < 0.05). *Treponema* showed a negative correlation with proline (*p* < 0.05). Prevotellaceae_UCG-001 showed a positive correlation with tryptophan (*p* < 0.05).

## 4. Discussion

### 4.1. Effect of Treatment on Milk Performance of Buffalo

The milk yield and milk quality of buffalo mainly depend on the intake of dry matter and the quality of feed, for which the quality of dietary protein and composition of the diet are the main influencing factors. It is well known that methionine is the first limiting amino acid in ruminant lactation. Rumen bypass protection of methionine can increase the absorption of amino acids in the small intestine and promote the synthesis of milk protein. Many experiments have shown that supplementing rumen-protected methionine can increase milk yield, milk protein content, and milk fat percentage [25,26]. Early studies [27] suggested that rumen bypass methionine could increase the yield and dry matter intake of dairy cows. However, some studies [28] believed that rumen-protected methionine increased the yield of true milk protein, with a slight decrease in dry matter intake and milk fat percentage, and a slight increase in milk production. The results of this study showed that palmitate coated L-methionine in the diet reduced the dry matter intake of buffaloes, but had no significant impact on the milk yield and milk quality of buffaloes. The reason may be due to different dietary structures and different types of rumen-protected methionine. In addition, composition of the diet is one of key factors which influence the milk yield and composition [29]. Sodium nitrate is often used as a methane inhibitor and non-protein nitrogen additive of ruminants, but excessive sodium nitrate may reduce food intake and cause nitrite poisoning. A study [30] found that when rumen microorganisms did not adapt to sodium nitrate, sodium nitrate would reduce the dry matter digestibility of forage animals, and after adaptation, sodium nitrate would not affect the dry matter digestibility. In addition, some in vitro studies of buffalo showed that sodium nitrate did not affect the dry matter digestibility [7]. The results of this study were similar to those of previous studies. The experiment lasted for 42 days. It was found that the addition of 70 g/d sodium nitrate to the diet did not affect the dry matter intake and milk production performance of buffaloes. The study also found that the combined use of sodium nitrate and methionine had no effect on the dry matter intake, milk yield, and milk composition of buffaloes. This may be because the buffalo has adapted to sodium nitrate. In addition, the limiting amino acids of lactating cows include lysine and methionine. The limiting sequence of lysine and methionine depends on the source of rumen protein [31].

### 4.2. Effect of Treatment on Rumen Fermentation Parameters

In the present study, the addition of methionine significantly reduced the content of acetate, propionate, valerate, and TVFA in the rumen of buffaloes, which may be related to the decrease in dry matter intake of buffaloes. Dietary fiber is the main source of rumen VFA. However, some studies have found that the addition of methionine in vitro has no effect on the content of single VFA and total VFA in buffalo rumen [13]. Moreover, supplementation of methionine under in vitro and in vivo studies in cattle showed no effect on rumen fermentation and N digestion [32]. Many studies [4,7,13,33] have shown that nitrate has toxic effects on rumen bacteria and has a strong inhibitory effect on the in vitro rumen fermentation process, which is shown by the reduction in VFA concentration. When the concentration of sodium nitrate added in vitro is more than 3 mg/mL, the concentration of TVFA in the rumen decreases significantly [9]. However, this study found that sodium nitrate had no significant effect on VFA concentration in buffalo rumen. The reason may be due to different dietary structures, different nitrate dosages, and adding methods. Proper dosage of sodium nitrate is beneficial to rumen fermentation and increased VFA concentration [9]. However, high doses of sodium nitrate may inhibit rumen fermentation, or cause poisoning and death of animals [15]. In addition, the effects of adding sodium nitrate through in vitro [4,7] and in vivo [34] methods on rumen fermentation are also different.

The rumen microbial protein mainly comes from the synthesis of microbial protein by rumen microorganisms. In this study, urea was used to balance the nitrogen content of each group of additives, and the influence of non-protein nitrogen content on rumen microbial protein synthesis was excluded. The study found that sodium nitrate and methionine had no significant effect on rumen microbial protein and ammonia nitrogen content. An in vitro study [35] conducted nitrogen equivalent treatment on feed non-protein nitrogen, and the results showed that the addition of sodium nitrate did not significantly affect rumen NH_3_-N and MCP, which was consistent with the results of this study.

### 4.3. Effect of Treatment on Ruminal Amino Acids

Rumen-free amino acid mainly reflects the degradation of dietary protein by rumen microorganisms. Previous in vitro studies [7] showed that the addition of sodium nitrate could increase the content of most essential amino acids and total amino acids in buffalo rumen, and had a positive impact on the composition of free amino acids in buffalo rumen. The results of this experiment also show that the addition of sodium nitrate has a greater impact on the composition of ruminal amino acids, which significantly increased the concentration of total and individual essential and non-essential amino acids as compared to control groups. These findings are consistent with the early research results [7]. The increase in dietary protein degradation rate and the inhibition of the process of amino acid decomposition into ammonia nitrogen will increase the rumen-free amino acid content [35,36]. It shows that sodium nitrate can promote the degradation of dietary protein in the rumen to free amino acids. In addition, sodium nitrate may decrease deamination of amino acids, but the mechanism remains to be studied.

In vitro studies [13] showed that in the presence of nitrate, the lower level of methionine (0.28%) had no effect on amino acid metabolism, but the higher level of methionine (1.12%) had a negative effect on amino acid content in the rumen. This study found that dietary methionine 15 g/d did not affect rumen amino acid metabolism. These results indicate that rumen microorganisms need the best concentration of amino acid, and amino acid above this concentration cannot promote fiber degradation, or can even have a negative impact on rumen fermentation.

### 4.4. Effect of Treatment on Rumen Microbial Community

The alpha diversity analysis of the buffalo rumen bacterial community in this study showed that the coverage rate of each group was higher than 98%, indicating that the sequencing results truly reflected the species and structural diversity of the buffalo rumen bacterial community. In addition, this study also found that sodium nitrate or methionine alone did not affect the alpha diversity index, but the combination of sodium nitrate and methionine could significantly improve Shannon, Ace, and Chao indexes. The results showed that the combination of sodium nitrate and methionine increased the diversity and richness of rumen microorganisms in buffalo. Patra et al. [37] found that sodium nitrate did not affect the Chao and Shannon indexes in vitro, which was consistent with the results of this test. In addition, some in vitro studies [13] found that the combined use of sodium nitrate and high-level methionine did not affect the Chao, Ace, and Shannon indexes, but increased the Simpson index, which was different from the results of this study. The reason may be different from the in vivo and in vitro test conditions.

This study found that Bacteroidota, Firmicutes, and Proteobacteria were the main bacterial phyla in the buffalo rumen, and *Prevotella* at the genus level, which is in agreement with earlier reports on rumen bacteria in buffalo [38]. Some studies [39] found that Bacteroidota played an important role in the degradation of non-fibrous substances, and the non-fibrous substances produced propionic acid after rumen fermentation. Spirochaetota can degrade cellulose, hemicellulose, pectin, etc., which has an important impact on the conversion of plant fibrous substances into VFA [40]. In addition, Proteobacteria also participates in the formation of biofilm and the digestion of soluble carbohydrates [41]. Actinobaciota plays an important role in the decomposition of cellulose. Our research shows that the combined use of nitrate and methionine increases the relative abundances of Proteobateria (including *Aconitobacter*, *Klebsiella*, and *Pseudomonas*), and Actinobaciota (including *Corynebacterium*), and decreases the relative abundances of Bacteroidota (including *Prevotell*, Prevotelaceae_UCG-001, Prevotelaceae_UCG-003, NK4A214_group, and norank_f__Backoidales_RF16_group) and Spirochaetota (including *Treponema*). Moreover, studies have shown that microbial communities can adapt to dietary nitrate by increasing the population of nitrate-reducing bacteria [42], which may be a possible reason as a substantial increase in the relative abundance of Proteobateria was observed in the SN+MET group. One study showed that the combined use of 1% sodium nitrate and 1.12% methionine could increase the relative abundance of *Compylobactor* in Proteobateria [13]. Our results showed that the combined use of nitrate and methionine inhibited the growth of microorganisms producing propionic acid, which also explained the reason why the concentration of propionate decreased significantly. However, the combined use of nitrate and methionine has no significant effect on the total VFA content, which indicates that the changes in VFA content and composition may be related to the degree of degradation of fibrous and non-fibrous materials by different microorganisms.

In this study, it was found that the addition of sodium nitrate alone increased the relative abundance of Rikenellaceae_ RC9_ gut_ group, and Rikenellaceae_ RC9_ gut_ group was identified as a biomarker of the sodium nitrate group in the rumen. However, sodium nitrate reduced the relative abundance of *Treponema* and did not affect other genera. Rikenellaceae is related to the degradation of mucin [43] and plays an important role in the health of intestinal mucosa. In addition, the relative abundance of Rikenellaceae is positively correlated with the feed utilization rate of the host and the metabolism of volatile fatty acids and short-chain fatty acids [44,45]. It is speculated that adding a proper amount of sodium nitrate may have a positive effect on the health of intestinal mucosa and the formation of VFA. In our study, we also found that *Treponema* was negatively correlated with proline. The sodium nitrate group reduced the relative abundance of *Treponema* and increased the content of proline. This proves that sodium nitrate is helpful in the synthesis of proline.

Our results showed that *Acinetobacter*, *Kurthia*, *Bacillus*, and *Corynebacterium* were identified as biomarkers of the sodium nitrate and methionine group in the rumen. In addition, *Acinetobacter*, *Kurthia*, *Bacillus*, and *Corynebacterium* showed a positive correlation with cysteine. *Acinetobacter* belongs to Proteus and has a strong ability to degrade cellulose [46]. *Kurthia* and *Bacillus* belong to Firmicutes. It has been reported that *Kurthia* has good stress resistance [47,48] and multiple degradation functions [49]. *Corynebacterium* can efficiently degrade cellulose and lignin [50]. The above shows that the combined use of nitrate and methionine can promote the growth of some cellulose-degrading bacteria and facilitate the formation of cysteine. However, in this study, the nitrate and methionine groups had no significant effect on the cysteamine content, and the reasons for this need to be further explored.

The correlation analysis between microorganisms and rumen fermentation showed that *Bacillus, Pedobacter*, *Klebsiella*, *Microbacterium*, *Chryseobacterium*, *Lactococcus*, and *Acinetobacter* were negatively correlated with acetate, propionate, valerate, and TVFA. The relative abundance of these genera increased significantly in the combination of nitrate and methionine. This is the reason why the concentration of propionic acid and valeric acid decreases.

## 5. Conclusions

Adding 70 g/d sodium nitrate to the diet increased the content of ruminal free amino acids of buffalo; 15 g/d methionine reduced the intake and rumen volatile fatty acids content of buffalo. The combined use of sodium nitrate and methionine enriched the species abundance of microorganisms in the rumen and increased the abundance of microorganisms positively correlated with cysteine and negatively correlated with acetate and propionate. However, sodium nitrate, methionine, and their combination had no significant effect on the milk yield and milk composition. The results showed that the combined use of sodium nitrate and methionine had a positive impact on the diversity of buffalo rumen microflora. It was suggested that the combined use of sodium nitrate and methionine was better in buffalo production.

## Figures and Tables

**Figure 1 microorganisms-11-00675-f001:**
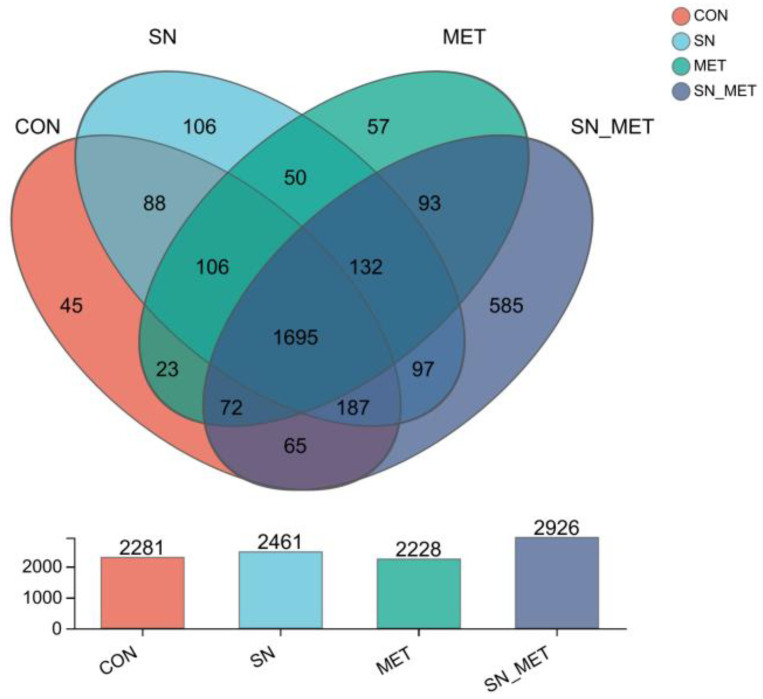
OTU distribution in the control (CON), 70 g/d sodium nitrate (SN), 15 g/d L-methionine (MET), and sodium nitrate group 70 g/d + L-methionine 15 g/d (SN+MET) groups.

**Figure 2 microorganisms-11-00675-f002:**
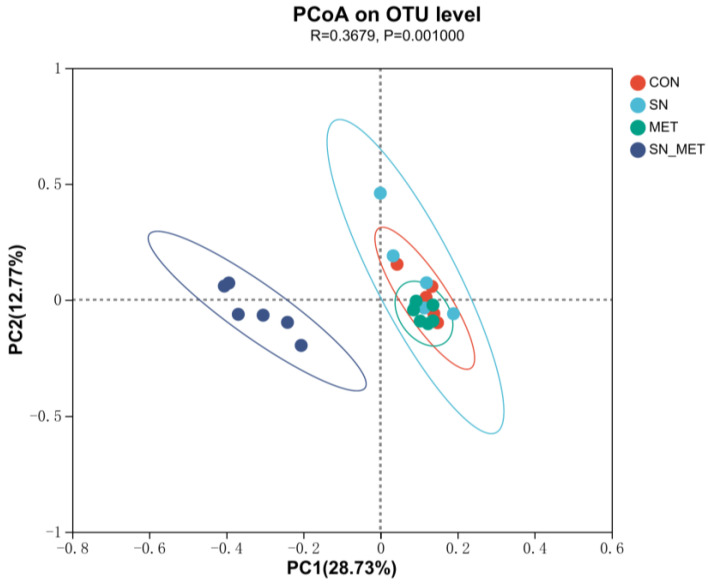
PCoA analysis of taxonomical classifications in the control (CON), 70 g/d sodium nitrate (SN), 15 g/d L-methionine (MET), and sodium nitrate group 70 g/d + L-methionine 15 g/d (SN+MET) groups.

**Figure 3 microorganisms-11-00675-f003:**
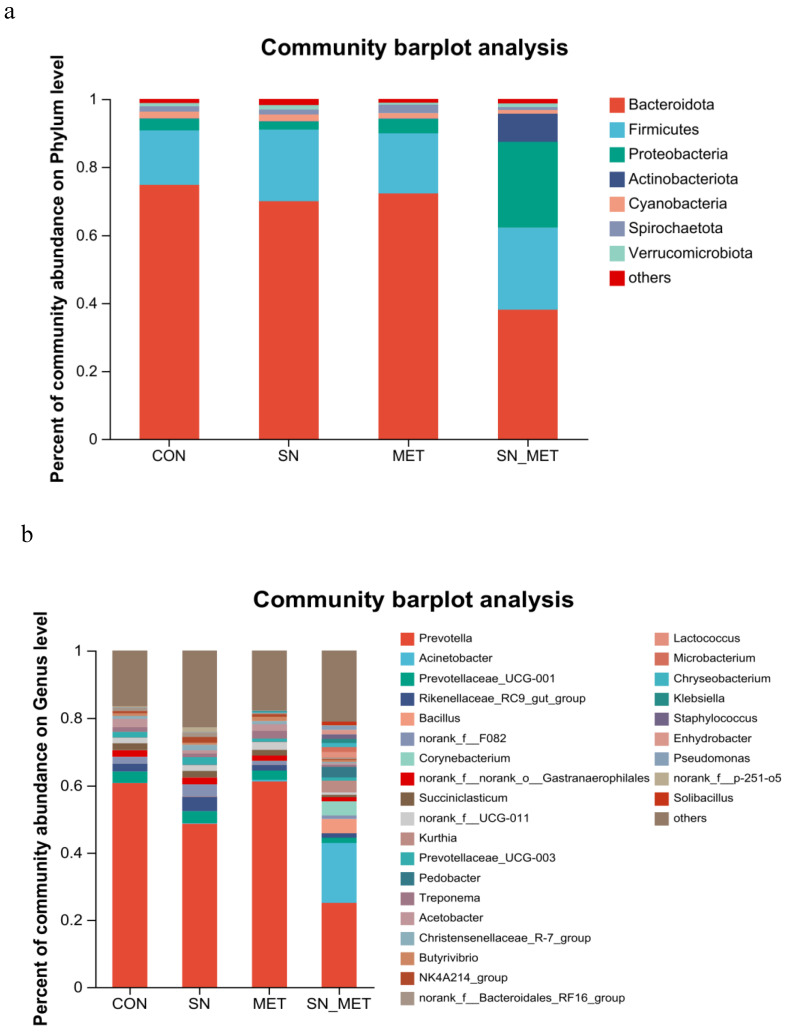
Relative abundance of rumen microflora of buffalo at the phylum level (**a**) and genus level (**b**) in the control (CON), 70 g/d sodium nitrate (SN), 15 g/d L-methionine (MET), and sodium nitrate group 70 g/d + L-methionine 15 g/d (SN+MET) groups.

**Figure 4 microorganisms-11-00675-f004:**
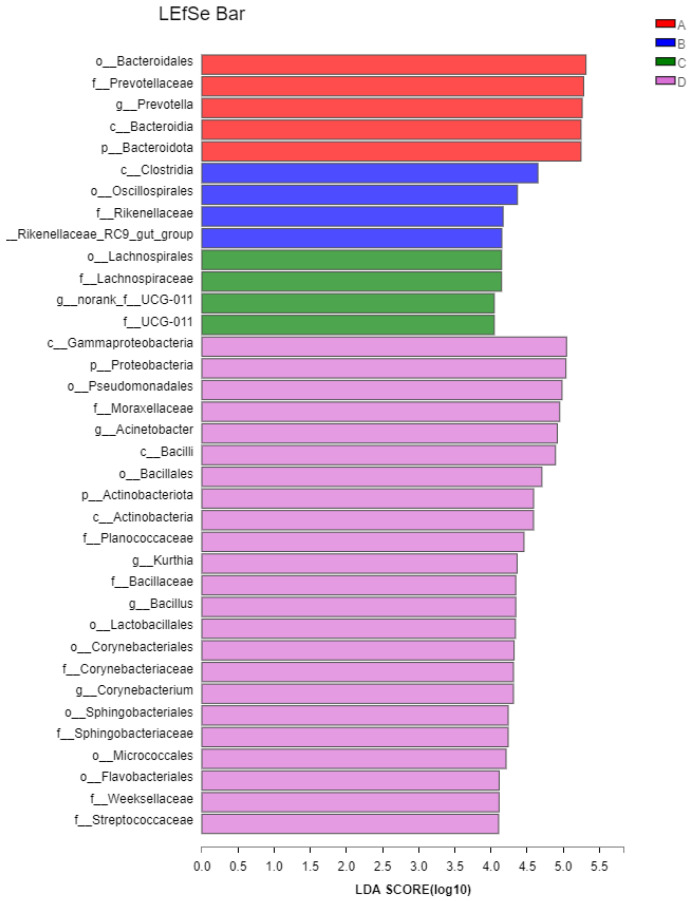
Biomarker bacterial genera in different treatment groups (the control (CON), 70 g/d sodium nitrate (SN), 15 g/d L-methionine (MET), and sodium nitrate group 70 g/d + L-methionine 15 g/d (SN+MET) groups) as revealed by linear discriminant analysis (LDA) Effect Size (LEfSe)-based analysis (LDA score> 4.0).

**Figure 5 microorganisms-11-00675-f005:**
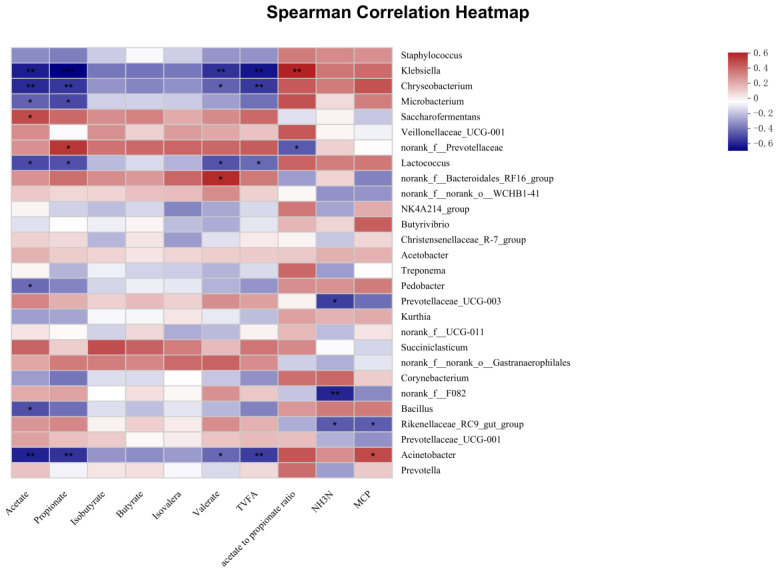
Correlation analyses between the fermentation characteristics and the top 28 rumen bacterial genera. In the two-dimensional heat map, the change in defined color and its depth indicates the nature and strength of the correlation, respectively. An asterisk sign was used when the r value was greater than 0.1 and the *p* values were less than 0.05 (* *p* < 0.05, ** *p* < 0.01, *** *p* < 0.001).

**Figure 6 microorganisms-11-00675-f006:**
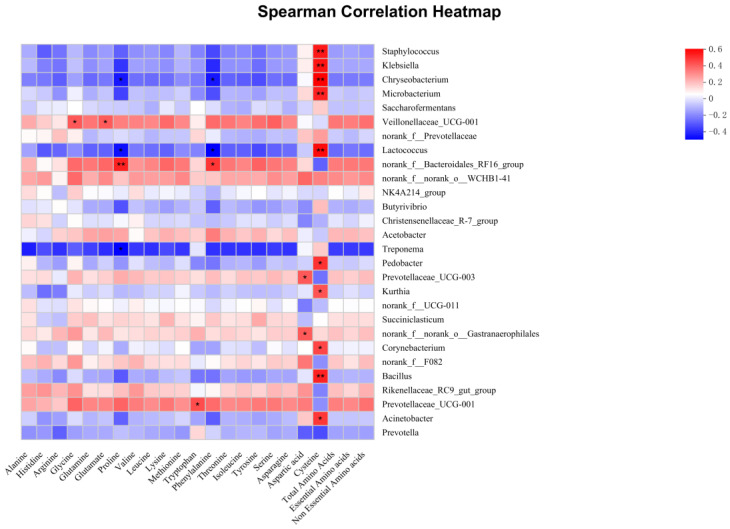
Correlation analyses between the amino acid contents and top 28 rumen bacterial genera. In the two-dimensional heat map, the change in defined color and its depth indicates the nature and strength of the correlation, respectively. An asterisk sign was used when the r value was greater than 0.1 and the *p* values were less than 0.05 (* *p* < 0.05, ** *p* < 0.01).

**Table 1 microorganisms-11-00675-t001:** Major ingredients and chemical composition of the experimental basal feed substrate based on dry matter (DM).

Items	Ingredient (%)			
	CON	SN	MET	SN+MET
Brewer’s grain	50.00	50.00	50.00	50.00
Corn leaf	16.00	16.00	16.00	16.00
Silage elephant grass	20.00	20.00	20.00	20.00
Corn	3.60	3.60	3.60	3.60
Soybean meal	7.10	7.10	7.10	7.10
Wheat bran	1.00	1.00	1.00	1.00
Cottonseed meal	1.25	1.25	1.25	1.25
CaHPO_4_	0.10	0.10	0.10	0.10
Shell meal	0.10	0.10	0.10	0.10
NaCl	0.15	0.15	0.15	0.15
NaHCO_3_	0.20	0.20	0.20	0.20
Premix ^1^	0.50	0.50	0.50	0.50
Urea (g)	26.41	1.51	24.89	50.00
Sodium nitrate (g)		70.00		70.00
L-methionine (g)			15.00	15.00
Chemical composition				
NE_L_ ^2^ (MJ/kg)	5.45	5.45	5.45	5.45
Crude protein (%)	20.75	20.75	20.75	20.75
Neutral detergent fiber (%)	50.61	50.61	50.61	50.61
Acid detergent Fiber (%)	21.56	21.56	21.56	21.56
Ash (%)	5.87	5.87	5.87	5.87
Ca (%)	0.97	0.97	0.97	0.97
P (%)	0.61	0.61	0.61	0.61

^1^ The additive premix provided the diet with the following (per kg of diet): Vitamin A 550,000 IU, Vitamin E 3000 IU, Vitamin D3 150,000 IU, 4.0 g Fe (as ferrous sulfate), 1.3 g Cu (as copper sulfate), 3.0 g Mn (as manganese sulfate), 6.0 g Zn (as zinc sulfate), 80 mg Co (as cobalt sulfate). ^2^ NE_L_: net energy for lactation. NE_L_(MJ/kg) = 0.703 × metabolic energy − 0.19.

**Table 2 microorganisms-11-00675-t002:** Effects of sodium nitrate and methionine on milk yield and milk composition (N = 10).

Items	CON	SN	MET	SN+MET	SEM	*p* Value
Dry matter intake (DMI, kg/d)	8.44 ^a^	8.35 ^a^	7.73 ^b^	8.17 ^a^	0.12	0.001
Milk yield (kg/d)	6.62	5.63	6.72	6.61	0.62	0.570
3.5% fat-corrected milk yield (3.5% FCM, kg/d)	3.17	2.72	3.22	3.18	0.29	0.585
Lactation efficiency (%)	78.86	67.50	87.54	80.43	7.82	0.342
Milk protein (%)	4.99	4.94	4.76	4.80	0.12	0.527
Milk fat (%)	8.43	8.55	8.37	8.86	0.31	0.695
Lactose (%)	5.18	5.11	5.39	5.34	0.08	0.060
Total solid content (%)	19.66	19.67	19.49	20.02	0.41	0.831
Sodium nitrate residue in milk (mg/kg)	3.72	2.13	2.96	2.96	0.45	0.113

Values with different superscripts in the same row differ significantly. CON was fed on a total mixed ratio (TMR), SN was fed on TMR supplemented with 70 g/d sodium nitrate, MET was fed on TMR supplemented with 15 g/d L-methionine, and SN+MET was fed on TMR supplemented with sodium nitrate group 70 g/d + L-methionine 15 g/d.

**Table 3 microorganisms-11-00675-t003:** Effects of sodium nitrate and methionine on rumen fermentation parameters (N = 10).

Items	CON	SN	MET	SN+MET	SEM	*p* Value
pH	6.48	6.63	6.64	6.73	0.08	0.208
Ammonia nitrogen (NH_3_-N, mg/mL)	2.03	1.55	2.36	1.75	0.59	0.790
Microbial protein (MCP, mg/mL)	0.65	0.63	0.77	0.77	0.06	0.169
Acetate (mmol/L)	35.29 ^a^	35.54 ^a^	29.69 ^b^	31.39 ^ab^	1.54	0.031
Propionate (mmol/L)	19.24 ^a^	18.68 ^a^	14.75 ^b^	14.95 ^b^	1.18	0.020
Isobutyrate (mmol/L)	0.67	0.72	0.59	0.61	0.06	0.485
Butyrate (mmol/L)	11.75	12.07	10.12	10.38	0.84	0.296
Isovalerate (mmol/L)	1.48	1.40	1.13	1.07	0.20	0.142
Valerate (mmol/L)	1.46 ^a^	1.62 ^a^	1.00 ^b^	0.85 ^b^	0.19	0.003
Total volatile fatty acids (TVFA, mmol/L)	69.90 ^a^	70.03 ^a^	57.29 ^b^	59.24 ^ab^	2.08	0.031
Acetate-to-propionate ratio (A/P)	1.86	1.92	2.02	2.14	0.09	0.154

Values with different superscripts in the same row differ significantly. CON was fed on a total mixed ratio (TMR), SN was fed on TMR supplemented with 70 g/d sodium nitrate, MET was fed on TMR supplemented with 15 g/d L-methionine, and SN+MET was fed on TMR supplemented with sodium nitrate group 70 g/d + L-methionine 15 g/d.

**Table 4 microorganisms-11-00675-t004:** Effects of sodium nitrate and methionine on rumen-free amino acids (N = 10) (µg/mL).

Items	CON	SN	MET	SN+MET	SEM	*p* Value
Alanine	21.56	29.63	28.42	13.75	5.30	0.161
Valine	7.54 ^ab^	13.74 ^a^	8.22 ^ab^	4.90 ^b^	2.05	0.041
Histidine	3.21 ^b^	5.39 ^a^	3.67 ^b^	2.86 ^b^	0.63	0.045
Arginine	1.04	1.24	1.06	0.83	0.12	0.152
Glycine	8.90	17.09	11.15	6.23	2.70	0.060
Glutamine	15.60 ^b^	35.372 ^a^	16.79 ^b^	12.90 ^b^	5.08	0.020
Glutamate	24.37 ^b^	47.18 ^a^	25.64 ^b^	17.19 ^b^	7.07	0.040
Proline	8.48 ^ab^	14.84 ^a^	6.17 ^ab^	4.54 ^b^	2.25	0.022
Leucine	7.59 ^b^	15.64 ^a^	7.77 ^b^	4.99 ^b^	2.33	0.024
Lysine	16.14 ^b^	35.37 ^a^	17.30 ^b^	12.80 ^b^	5.37	0.032
Methionine	3.84 ^b^	7.94 ^a^	4.36 ^b^	2.89 ^b^	1.15	0.029
Tryptophan	0.67	1.21	0.62	0.83	0.21	0.223
Phenylalanine	6.71 ^b^	12.68 ^a^	5.12 ^b^	3.53 ^b^	1.84	0.012
Threonine	8.35 ^b^	16.84 ^a^	8.70 ^b^	5.61 ^b^	2.38	0.019
Isoleucine	7.46 ^b^	14.93 ^a^	7.54 ^b^	4.80 ^b^	2.23	0.026
Tyrosine	4.29 ^b^	9.04 ^a^	4.26 ^b^	3.02 ^b^	1.34	0.023
Serine	8.55 ^b^	14.74 ^a^	8.56 ^b^	5.33 ^b^	2.17	0.042
Aspartic acid	7.05 ^b^	13.79 ^a^	7.27 ^b^	4.9 ^b^	2.09	0.038
Asparagine	0.90	1.09	0.91	0.98	0.06	0.096
Cysteine	0.74	0.73	1.50	1.34	0.24	0.065
Essential amino acids ^1^	58.30 ^b^	118.35 ^a^	53.97 ^b^	47.14 ^b^	10.13	0.032
Non-essential amino acids ^2^	104.67 ^ab^	190.13 ^a^	103.87 ^ab^	88.03 ^b^	15.67	0.071
Total amino acids	162.99 ^b^	308.48 ^a^	175.03 ^b^	114.2 ^b^	44.81	0.036

Values with different superscripts in the same row differ significantly. CON was fed on a total mixed ratio (TMR), SN was fed on TMR supplemented with 70 g/d sodium nitrate, MET was fed on TMR supplemented with 15 g/d L-methionine, SN+MET was fed on TMR supplemented with sodium nitrate group 70 g/d + L-methionine 15 g/d. ^1^ Isoleucine, leucine, lysine, methionine, phenylalanine, threonine, tryptophan, valine; ^2^ Histidine, alanine, arginine, glycine, glutamine, glutamate, proline, tyrosine, serine, aspartic acid, asparagine, cysteine.

**Table 5 microorganisms-11-00675-t005:** Alpha diversity parameters across different treatment groups.

Items	CON	SN	MET	SN+MET	SEM	*p* Value
Shannon	5.18	5.38	5.24	4.86	0.16	0.161
Simpson	0.02 ^b^	0.02 ^b^	0.02 ^b^	0.04 ^a^	0.01	0.025
Ace	1611.18 ^b^	1768.28 ^b^	1658.22 ^b^	2121.93 ^a^	74.94	0.001
Chao	1591.28 ^b^	1777.19 ^b^	1668.88 ^b^	2126.42 ^a^	80.47	0.001
Coverage (%)	98.89	98.97	99.09	99.04	0.06	0.674

Values with different superscripts in the same row differ significantly. CON was fed on a total mixed ratio (TMR), SN was fed on TMR supplemented with 70 g/d sodium nitrate, MET was fed on TMR supplemented with 15 g/d L-methionine, and SN+MET was fed on TMR supplemented with sodium nitrate group 70 g/d + L-methionine 15 g/d.

## Data Availability

The sequence data generated in this experiment (16SrRNA gene sequences) were deposited in SRA database of NCBI under Bioproject No. PRJNA912909.

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
