# Peer review of "Effects of Sodium Nitrate and Coated Methionine on Lactation Performance, Rumen Fermentation Characteristics, Amino Acid Metabolism, and Microbial Communities in Lactating Buffaloes"

_microorganisms, 2023, doi:10.3390/microorganisms11030675_

Round 1
Reviewer 1 Report
I would like to suggest that you do recheck your grammatical mistake using software.
Line 4: Changed to microbial communities. Changed to buffaloes
Line 12: Please give introduction
Line 14: Changed to microbial communities. Changed to buffaloes
Line 14: Give the information of DIM (day in milk), and how many animals are in primiparous and multiparous
Line 15-17: All of animals received the same TMR. Furthermore, the group were divided into….. etc.
Line 18-28: Although no significant, please give the information about milk yield and milk protein.
Line 42: give the reference after stable.
Line 45: Delete and
Line 48: Delete further proving this view
Line 49: Put Furthermore, before palmitate
Line 49-55 and 58-60: In order to make the story is clear, please change the sentences
Line 60: please give the introduction also about sodium nitrate.
Line 63-65: I don’t understand your story, please make it clear
Line 71-73: Please give the information about DIM (day in milk), and how many animals are in primiparous and multiparous
Line 77-80 : Which feeding standard did you follow? Please give the information and references.
Line 85: Please give the info how did you calculate NEL
Line 87: Why only 3 days ?
Line 89-91: Please reorganize the sentence.
Line 95: (Foss Electric, Hildesforth Electric Company, Denmark), including protein, etc….
Line 103: Why only measure 1 day?
Line 104-105: Please reorganize the sentence
Line 118: When did you take the samples? In which day? And how many samples of x animals
Line 128: When did you take the samples? In which day? And how many samples of x animals
Line 200: OUT
Line 297-300: Please give the detailed information, what might be the reason behind? Because there was difference in DMI but no effect in milk yield and milk protein.
Line 313-319: Please give the info about maximum dosage which could impair rumen fermentation
Line 350-353: Please give the info about maximum dosage which could impair amino acids
Line 354: How D treatment could hamper the population of Bacteroidota ? and increase Proteobacteria and actinobacteria? Please give the reason (more references).
Author Response
Dear reviewer:
First, we would like to thank you for the positive and constructive comments and suggestions.
All authors have read and approved the manuscript. All necessary corrections have been made, which can be viewed as track changes. The point to point responses for each comment are as follows:
Reviewer #1:
Comments and Suggestions for Authors
1.I would like to suggest that you do recheck your grammatical mistake using software.
Answer: We have rechecked and corrected grammatical mistake using software.
2.Line 4: Changed to microbial communities. Changed to buffaloes.
Answer: We have changed. Please see line 4.
3.Line 12: Please give introduction.
Answer: We have given introduction of sodium nitrate and methionine as suggested. Please see lines 14-15.
4.Line 14: Changed to microbial communities. Changed to buffaloes.
Answer: We have changed. Please see line 18.
5.Line 14: Give the information of DIM (day in milk), and how many animals are in primiparous and multiparous.
Answer: We have given the information. Please see lines 18-20.
6.Line 15-17: All of animals received the same TMR. Furthermore, the group were divided into….. etc.
Answer: We have revised it according to your suggestions. Please see lines 21-24.
- Line 18-28: Although no significant, please give the information about milk yield and milk protein.
Answer: We have given the information. Please see lines 28-30.
- Line 42: give the reference after stable.
Answer: We have given the reference after stable. Please see lines 55.
- Line 45: Delete and.
Answer: We have deleted “and”. Please see line 59.
- Line 48: Delete further proving this view.
Answer: We have deleted “further proving this view”. Please see line 61.
- Line 49: Put Furthermore, before palmitate.
Answer: We have revised. Please see line 62.
- Line 49-55 and 58-60: In order to make the story is clear, please change the sentences.
Answer: We have reorganized the sentences. Please see lines 62-67.
- Line 60: please give the introduction also about sodium nitrate.
Answer: We have given the introduction about sodium nitrate. Please see lines 70-73.
- Line 63-65: I don’t understand your story, please make it clear.
Answer: We have reorganized the sentences. Please see lines 73-76.
- Line 71-73: Please give the information about DIM (day in milk), and how many animals are in primiparous and multiparous.
Answer: We have given the information. Please see lines 86-87.
- Line 77-80 : Which feeding standard did you follow? Please give the information and references.
Answer: The nutritional level of the experimental diet meets NRC (2001). Please see line 96.
- Line 85: Please give the info how did you calculate NEL.
Answer:We have provided the calculation formula of NEL. Please see line 105.
- Line 87: Why only 3 days ?
Answer: Sorry, I didn't describe it clearly enough. The dry matter intake (DMI) was measured every day during the formal period. The DMI was calculated daily by the TMR and refusals every day. Please see lines 107-108.
- Line 89-91: Please reorganize the sentence.
Answer: We have reorganized the sentence. Please see lines 108-113.
- Line 95: (Foss Electric, Hildesforth Electric Company, Denmark), including protein, etc….
Answer: We have revised it according to your suggestions. Please see line 117.
- Line 103: Why only measure 1 day?
Answer: Rumen fluid was collected and measured for one day. This measurement method has been widely reported. Please see articles for reference ( Li, Y.; Wang, J.; Mei, J.; Huang, L.; Liu, H. Effects of Mulberry Branch and Leaves Silage on Microbial Community, Rumen Fermentation Characteristics, and Milk Yield in Lactating Dairy Cows. Fermentation 2022, 8, 86. https://doi.org/10.3390/fermentation8020086; Shen, J.; Chai, Z.; Song, L.; Liu, J.; Wu, Y. Insertion depth of oral stomach tubes may affect the fermentation parameters of ruminal fluid collected in dairy cows. J. Dairy Sci. 2012, 95, 5978–5984. ).
- Line 104-105: Please reorganize the sentence.
Answer: We have reorganized the sentence. Please see lines 128-130.
- Line 118: When did you take the samples? In which day? And how many samples of x animals.
Answer: The rumen content samples (40) collected on the last day of the experiment were taken out. Please see line144.
- Line 128: When did you take the samples? In which day? And how many samples of x animals.
Answer: The rumen content samples (40) collected on the last day of the experiment were taken out. Please see line155.
- Line 200: OUT.
Answer: We have corrected it. Please see line 230.
- Line 297-300: Please give the detailed information, what might be the reason behind? Because there was difference in DMI but no effect in milk yield and milk protein.
Answer: We have given the detailed information. Please see lines 345-346.
- Line 313-319: Please give the info about maximum dosage which could impair rumen fermentation.
Answer: We have given the detailed information. Please see lines 372-373.
- Line 350-353: Please give the info about maximum dosage which could impair amino acids.
Answer: The maximum dosage of methionine impairing amino acids is in the line 406. Sorry, we can't find the reference literature about the maximum dose of sodium nitrate that impairs amino acids.
- Line 354: How D treatment could hamper the population of Bacteroidota ? and increase Proteobacteria and actinobacteria? Please give the reason (more references).
Answer: We have given the reason according to your suggestions. Please see lines 440-445.
Reviewer 2 Report
General remarks:
The topic of this manuscript is interesting for Microorganisms’ readers, but I find many problematic parts in the text.
Details:
In the whole text: in vivo and in vitro phrases please indicate italic style!
line 13: not lactation performance! Rather daily milk yield and milk composition!
line 34: see line 13!
lines 54-55: This sentence is not clear! Please rewrite this sentence!
line 61: “nitric oxide”: previously not found any information! Please clarify it!
lines 61-62: “animal liver”: suggest only liver and add the species in brackets!
line 71: parity? Please add data of DIM (days in milk)!
Table 1: please add Nel in MJ/kg!
lines 83-84: please add a “vitamin” phrase instead of “V”!
line 86: DMI instead of Dmi
Table 2: title of table: why not use abbreviation of treatments? E.g. C or TMR as control, TMR +SN; TMR+M; TMR+SN+M?
Table 2,3 and 4: please add the number of elements in the Table!
Table 3: total volatile fatty acids!
Figure 1: title of the Figure: please correct the title (“OUT”)!
line 286: not only quality, but composition of the diet is one of key factors which influence the milk yield and composition! Please clarify it!
lines 287-300: please discuss the effect of the unprotected methionine on milk parameters!
lines 284-307: please discuss the combined effect of Met and sodium nitrate on milk parameters!
line 320: “…adding methods.” Please clarify it!
lines 324-325: urea was used in the experiment, but no data in Table 1 and M&M section!!!
line 422: volatile fatty acids!
line 427: see line 13!
Author Response
Dear reviewer:
First, we would like to thank you for the positive and constructive comments and suggestions.
All authors have read and approved the manuscript. All necessary corrections have been made, which can be viewed as track changes. The point to point responses for each comment are as follows:
Reviewer #2:
Comments and Suggestions for Authors
General remarks:
The topic of this manuscript is interesting for Microorganisms’ readers, but I find many problematic parts in the text.
Details:
- In the whole text: in vivo and in vitro phrases please indicate italic style!
Answer: All "in vivo" and "in vitro" have been changed to italics.
2.line 13: not lactation performance! Rather daily milk yield and milk composition!
Answer: We have changed “lactation performance” to “ milk yield and milk composition”. Please see line 17.
3.line 34: see line 13!
Answer: We have changed “milk performance of buffaloes” to “ milk yield and milk composition”. Please see line 46.
4.lines 54-55: This sentence is not clear! Please rewrite this sentence!
Answer: We have rewrited this sentence. Please see lines 64-65.
5.line 61: “nitric oxide”: previously not found any information! Please clarify it!
Answer: We have added relevant information. Please see lines 70-73.
6.lines 61-62: “animal liver”: suggest only liver and add the species in brackets!
Answer: We have revised it according to your suggestions. Please see line 74.
7.line 71: parity? Please add data of DIM (days in milk)!
Answer: We have added data of DIM (days in milk). Please see lines 86-87.
8.Table 1: please add Nel in MJ/kg!
Answer: We have converted the unit of NEL Mcal/kg into MJ/kg. Please see Table 1.
9.lines 83-84: please add a “vitamin” phrase instead of “V”!
Answer: We have added a “vitamin” phrase instead of “V”. Please see lines 102-103.
10.line 86: DMI instead of Dmi .
Answer: We have revised it. Please see line 106.
11.Table 2: title of table: why not use abbreviation of treatments? E.g. C or TMR as control, TMR +SN; TMR+M; TMR+SN+M?
Answer: We have revised it according to your suggestions. CON, SN, MET, SN+MET instead of A, B, C, D. The whole manuscript has been corrected accordingly.
12.Table 2,3 and 4: please add the number of elements in the Table!
Answer: We have added the number of elements in the Table 2, 3 and 4.
13.Table 3: total volatile fatty acids!
Answer: We have revised it. Please see Table 3.
14.Figure 1: title of the Figure: please correct the title (“OUT”)!
Answer: We have corrected OUT to OTU. Please see line 230.
15.line 286: not only quality, but composition of the diet is one of key factors which influence the milk yield and composition! Please clarify it!
Answer: We have revised it according to your suggestions. Please see lines 332-333.
16.lines 287-300: please discuss the effect of the unprotected methionine on milk parameters!
Answer: Rumen-protected methionine (palmitate coated L-methionine) was used in our study. Ruminant feed is mainly supplemented with rumen-protected methionine. We have not found any references for unprotected methionine.
17.lines 284-307: please discuss the combined effect of Met and sodium nitrate on milk parameters!
Answer: We have added relevant information according to your suggestions. Please see lines 356-361.
18.line 320: “…adding methods.” Please clarify it!
Answer: We have revised it according to your suggestions. Please see lines 376-380.
19.lines 324-325: urea was used in the experiment, but no data in Table 1 and M&M section!!! Answer: We have added relevant information. Please see lines 94-95 and Table 1.
20.line 422: volatile fatty acids!
Answer: We have revised it. Please see line 485.
21.line 427: see line 13!
Answer: We have changed “milk performance of buffaloes” to “ milk yield and milk composition”. Please see line 489.
Reviewer 3 Report
This manuscript describes the results of NPN and methionine on milk yield and composition and rumen microbiome in lactating buffalo. The project appears to be well planned, conducted, and analyzed. My greatest concern is with the sampling procedures, which I will outline in the specific comments below. Overall good paper.
specific comments:
L14 - in the methods you use the term Murrah
L17 - is the methionine rumen protected?
L22 - define ACE
L71 - provide mean +/- SD for day sin milk at start of adaptation period
L73 - using Group A, B, C, D is fairly easy to follow but it might be easier for the reader if you used acronyms that provided some indication of the treatment such as CON, NO3, MET, NO3+MET
L75 - is the methionine rumen protected
L79 - Did the treatment diets start at parturition for each buffalo or were all 40 buffalo started on treatments on the same day (i.e. different days in milk)
Table 1 - since treatment diets were formulated to be isonitrogenous, ingredient composition for all 4 diets should be reported showing the different amounts of NaNO3 and Urea used.
L87 - Was DMI only measured for 3 days at the end of the experiment? This is not enough data to accurately determine DMI.
L89 - What is the prefeeding period? Is that the same as the adaptation period? Use the same term to be clear. Was milk yield only measured 1 day at the start of the adaptation period? This is not enough data to accurately assess milk yield.
L104 - How was rumen fluid collected? Stomach tube? Was any solid feed material collected? 75% of rumen microbes reside attached to feed particles.
L105 - why were buffalo held off feed prior to rumen sampling? How might this impact the results of microbiome?
L151 - include the experimental design - I assume a completely randomized design with only treatment as fixed effect.
Table 2 - should you include efficiency (milk per DMI), but if milk yield and DMI were measured at different times, then if may not be appropriate.
L175 - change capitalization on 'The'
L176 - include that Group C was not different than Group D
Table 4 - does the increased total amino acids of Group B indicate that rumen N balance was negative in Group A? But Group D was not increased, so does it mean that NO3 resulted in greater amino acid synthesis or less deamination?
Figure 1 - change 'OUT' to 'OTU'. Define A,B,C,D in the caption
Table 5 - define ABCD groups in footnote
Figure 2 - define ABCD in caption
Figure 3 - define ABCD in caption
Table 6 - this table is repeat of data in Figure 3, so may think about making this table supplemental table, But do not delete because I think it is important to report actual values and is easier to show statistical significance.
Figure 4 - define ABCD in caption.
L247-253 - It looks like there are more than 1, 1, 1, and 4 biomarkers with LDA >4 for A,B,C, and D in Figure 4
L272-277 - change 'negatively correlated' to 'negative correlation' and 'positively correlated' to 'positive correlation'
Figure 5 and 6 - In the caption define what *, **, and *** mean
L291,294 - what does 'milk fat rate' mean? I do not understand the 'rate' term. Do you mean concentration?
L298 - it appears in the methods that you did not use a rumen-protected methionine source
L300 - change to ' when rumen microorganisms'
L303 and throughout - use 'in vitro' and 'in vivo' as lowercase italicized
L304 - I do not know what digestibility of buffalo rumen means. Do you mean digestibility of some feedstuff in buffalo rumen? Please clarify.
L319 - expand this thought to discuss differences in diet nitrate dosages, and adding methods. Currently, this sentences leaves the reader hanging wondering exactly what you mean
L321 - rumen microorganisms do not synthesize dietary protein, they synthesize microbial protein. please reword the sentence
L342 - NH3-N was not lower with a P-value of 0.79
L344-346 - this is intriguing concept. This thought should be expanded to explain the potential mechanisms where NO3 decreases deamination of amino acids
L350 - change to 'affect rumen'
L358 - change to 'affect the alpha'
L367 - you use test, study, and experiment in different places. please use a single term throughout
L369 - do you mean genetic level?
L405 - Blautia is not mentioned in the list at start of this paragraph
L408-413 - If NO3 and Methionine can promote growht of cellulose-degrading bacteria then why does D have lower VFA indicating lower fiber digestion. in paragraph above you indicate that D had lower VFA indicating lower digestion and again in the next paragraph. I can see a shift in microbial population change the VFA profile while also increasing cellulose-degrading bacteria, but it is hard to explain decreasing total VFA while also increasing -cellulose-degrading bacteria. This needs further explanation.
Conclusion - the conclusion is just restating the results. It should be providing the meaning of the results and the take-home message.
Author Response
Dear reviewer:
First, we would like to thank you for the positive and constructive comments and suggestions.
All authors have read and approved the manuscript. All necessary corrections have been made, which can be viewed as track changes. The point to point responses for each comment are as follows:
Reviewer #3:
Comments and Suggestions for Authors
This manuscript describes the results of NPN and methionine on milk yield and composition and rumen microbiome in lactating buffalo. The project appears to be well planned, conducted, and analyzed. My greatest concern is with the sampling procedures, which I will outline in the specific comments below. Overall good paper.
specific comments:
1.L14 - in the methods you use the term Murrah.
Answer: We have corrected Mora to Murrah. Please see line 19.
2.L17 - is the methionine rumen protected?
Answer: Sorry, I didn't describe it clearly. Yes, palmitate coated L-methionine. The whole manuscript has been corrected accordingly.
3.L22 - define ACE.
Answer: I'm sorry for spelling mistakes. We have corrected ACE to Ace.
4.L71 - provide mean +/- SD for day sin milk at start of adaptation period .
Answer: We have added data of DIM (days in milk). Please see lines 86-87.
5.L73 - using Group A, B, C, D is fairly easy to follow but it might be easier for the reader if you used acronyms that provided some indication of the treatment such as CON, NO3, MET, NO3+MET .
Answer: We have revised it according to your suggestions. CON, SN, MET, SN+MET instead of A, B, C, D.
6.L75 - is the methionine rumen protected .
Answer: Yes, palmitate coated L-methionine. The whole manuscript has been corrected accordingly.
7.L79 - Did the treatment diets start at parturition for each buffalo or were all 40 buffalo started on treatments on the same day (i.e. different days in milk) .
Answer: All 40 buffaloes started on treatments on the same day. Average days in milk (DIM) = 180.83±56.7 d. Please see line 87.
8.Table 1 - since treatment diets were formulated to be isonitrogenous, ingredient composition for all 4 diets should be reported showing the different amounts of NaNO3 and Urea used.
Answer: We have revised it according to your suggestions and added relevant information. Please see lines 94-95 and Table 1.
9.L87 - Was DMI only measured for 3 days at the end of the experiment? This is not enough data to accurately determine DMI.
Answer: Sorry, I didn't describe it clearly enough. The dry matter intake (DMI) was measured every day during the formal period. The DMI was calculated daily by the TMR and refusals every day. Please see lines 107-108.
10.L89 - What is the prefeeding period? Is that the same as the adaptation period? Use the same term to be clear. Was milk yield only measured 1 day at the start of the adaptation period? This is not enough data to accurately assess milk yield.
Answer: We have reorganized the sentence. Please see lines 108-113.
11.L104 - How was rumen fluid collected? Stomach tube? Was any solid feed material collected? 75% of rumen microbes reside attached to feed particles.
Answer: Rumen content samples (500mL, both solid and fluid phase) were collected only once, at the last day of the experiment before the morning feeding, using a stomach tube. After collection, the samples were directly transported to the laboratory. Please see lines 128-130.
12.L105 - why were buffalo held off feed prior to rumen sampling? How might this impact the results of microbiome?
Answer: If the rumen content sample of the buffalo is collected with a stomach tube immediately after feeding, it may cause physical discomfort of the buffalo, such as vomiting. It may also be difficult to insert the stomach tube due to too much rumen feed. Therefore, samples of rumen contents were collected before feeding. Moreover, the method of collecting rumen content samples before feeding has been widely used in other studies. Please see articles for reference ( Hassan F U , Ebeid H M , Tang Z , et al. Citation: A Mixed Phytogenic Modulates the Rumen Bacteria Composition and Milk Fatty Acid Profile of Water Buffaloes. Frontiers in Veterinary Science, 2020, 7:569; Biscarini F , Palazzo F , Castellani F , et al. Rumen microbiome in dairy calves fed copper and grape-pomace dietary supplementations: Composition and predicted functional profile. PLoS ONE, 2018, 13(11)).
13.L151 - include the experimental design - I assume a completely randomized design with only treatment as fixed effect.
Answer: The experimental design of completely randomized design with only fixed treatment effect was adopted. Please see lines 179-180.
14.Table 2 - should you include efficiency (milk per DMI), but if milk yield and DMI were measured at different times, then if may not be appropriate.
Answer: We have added the data of lactation efficiency in Table 2 and the calculation formula of lactation efficiency in lines 119-120.
15.L175 - change capitalization on 'The' .
Answer: We have corrected “The” to “the”. Please see line 204.
16.L176 - include that Group C was not different than Group D.
Answer: We have added relevant information. There was no significant difference in all rumen fermentation parameters in Groups MET and SN+MET (p>0.05). Please see lines 207-208.
17.Table 4 - does the increased total amino acids of Group B indicate that rumen N balance was negative in Group A? But Group D was not increased, so does it mean that NO3 resulted in greater amino acid synthesis or less deamination?
Answer: When the buffalo is fasting, the free amino acids and ammonia nitrogen in the rumen fluid are utilized by the rumen microorganisms, and are not directly affected by additives and feed. Because the utilization of free amino acids by rumen microorganisms in group B (SN) is relatively low, the content of free amino acids in group B (SN) is higher than that in other groups.
18.Figure 1 - change 'OUT' to 'OTU'. Define A,B,C,D in the caption.
Answer: We have changed “OUT” to “OTU” and defined CON, SN, MET, SN+MET in the caption. Please see lines 230-231.
19.Table 5 - define ABCD groups in footnote .
Answer: We have defined CON, SN, MET, SN+MET groups in footnote. Please see Table 5.
20.Figure 2 - define ABCD in caption .
Answer: We have defined CON, SN, MET, SN+MET groups in footnote. Please see Figure 2.
21.Figure 3 - define ABCD in caption.
Answer: We have defined CON, SN, MET, SN+MET groups in footnote. Please see Figure 3.
22.Table 6 - this table is repeat of data in Figure 3, so may think about making this table supplemental table, But do not delete because I think it is important to report actual values and is easier to show statistical significance.
Answer: We have changed “Table 6” to “Table S1”. The Table S1 is uploaded as supplementary material.
23.Figure 4 - define ABCD in caption.
Answer: We have defined CON, SN, MET, SN+MET groups in footnote. Please see Figure 4.
24.L247-253 - It looks like there are more than 1, 1, 1, and 4 biomarkers with LDA >4 for A,B,C, and D in Figure 4.
Answer: We selected biomarkers at the genus level.
25.L272-277 - change 'negatively correlated' to 'negative correlation' and 'positively correlated' to 'positive correlation' .
Answer: We have changed 'negatively correlated' to 'negative correlation' and 'positively correlated' to 'positive correlation' . Please see lines 313-318.
26.Figure 5 and 6 - In the caption define what *, **, and *** mean.
Answer: We have defined *, **, and *** mean. Please see Figure 5 and 6.
27.L291,294 - what does 'milk fat rate' mean? I do not understand the 'rate' term. Do you mean concentration?
Answer: We have changed “milk fat rate” to “milk fat percentage”. Please see lines 337, 341.
28.L298 - it appears in the methods that you did not use a rumen-protected methionine source. Answer: Palmitate coated L-methionine was used in our experiment and has been modified in Materials and Methods.
29.L300 - change to ' when rumen microorganisms' .
Answer: We have changed to ' when rumen microorganisms'. Please see line 349.
30.L303 and throughout - use 'in vitro' and 'in vivo' as lowercase italicized.
Answer: All "in vivo" and "in vitro" have been changed to italics.
31.L304 - I do not know what digestibility of buffalo rumen means. Do you mean digestibility of some feedstuff in buffalo rumen? Please clarify.
Answer: It is the dry matter digestibility. We have revised it. Please see line 350.
32.L319 - expand this thought to discuss differences in diet nitrate dosages, and adding methods. Currently, this sentences leaves the reader hanging wondering exactly what you mean .
Answer: We have revised it according to your suggestions. Please see lines 376-380.
33.L321 - rumen microorganisms do not synthesize dietary protein, they synthesize microbial protein. please reword the sentence .
Answer: We have revised it according to your suggestions. Please see lines 381-382.
34.L342 - NH3-N was not lower with a P-value of 0.79 .
Answer: We have deleted these sentences. Please see line 401.
35.L344-346 - this is intriguing concept. This thought should be expanded to explain the potential mechanisms where NO3 decreases deamination of amino acids .
Answer: We have revised it according to your suggestions. Please see lines 402-403.
36.L350 - change to 'affect rumen' .
Answer: We have changed to 'affect rumen'. Please see line 407.
37.L358 - change to 'affect the alpha' .
Answer: We have changed to 'affect the alpha'. Please see line 416.
38.L367 - you use test, study, and experiment in different places. please use a single term throughout .
Answer: We have revised it according to your suggestions. We have changed to “study” throughout.
39.L369 - do you mean genetic level?
Answer: We have changed to 'genus leve'. Please see line 427.
40.L405 - Blautia is not mentioned in the list at start of this paragraph.
Answer: We have deleted “Studies have shown that Blautia is positively related to the relative abundance of amino acid biosynthesis, which can promote the biosynthesis of rumen-branched-chain amino acids [41].”
41.L408-413 - If NO3 and Methionine can promote growht of cellulose-degrading bacteria then why does D have lower VFA indicating lower fiber digestion. in paragraph above you indicate that D had lower VFA indicating lower digestion and again in the next paragraph. I can see a shift in microbial population change the VFA profile while also increasing cellulose-degrading bacteria, but it is hard to explain decreasing total VFA while also increasing -cellulose-degrading bacteria. This needs further explanation.
Answer: The relative abundance of some cellulose-degrading bacteria in Group D (SN+MET) was significantly higher than that in Group A (CON), but there was no significant difference in the concentration of TVFA between Group A (CON) and Group D (SN+MET) (Table 2). The combined use of nitrate and methionine has no significant effect on the total VFA content, which indicates that the changes in VFA content and composition may be related to the degree of degradation of fibrous and non-fibrous materials by different microorganisms.
42.Conclusion - the conclusion is just restating the results. It should be providing the meaning of the results and the take-home message.
Answer: Conclusion has been modified and provided the meaning of the results and the take-home message. Please see lines 489-492.
Reviewer 4 Report
Effects of sodium nitrate and methionine on lactation performance, rumen fermentation characteristics, amino acid metabolism, and microbial community in lactating buffalo
The manuscript is well-written. The hypothesis was well designed by the instructor, and the work is novel and unique. The test's power is normal, and the statistics are correct. M&M provided sufficient information. The authors should pay more attention to text formatting in accordance with journal guidelines. For example, all "in vivo" and "in vitro" wording (for example, L314,L347, etc.) should be italicized, and all subheadings (Discussion section) should be capitalized. Aside from formatting, the results were presented in full detail, and the obtained results were thoroughly discussed. Only the conclusion lacks a take-home message and simply repeats the findings. Aside from the existing sentences, the conclusion should provide a single clear message followed by a suggestion for future research.
I suggest author to revise their manuscript and re-submit it after minor revisions.
Author Response
Comments and Suggestions for Authors
Effects of sodium nitrate and methionine on lactation performance, rumen fermentation characteristics, amino acid metabolism, and microbial community in lactating buffalo
The manuscript is well-written. The hypothesis was well designed by the instructor, and the work is novel and unique. The test's power is normal, and the statistics are correct. M&M provided sufficient information. The authors should pay more attention to text formatting in accordance with journal guidelines. For example, all "in vivo" and "in vitro" wording (for example, L314,L347, etc.) should be italicized, and all subheadings (Discussion section) should be capitalized. Aside from formatting, the results were presented in full detail, and the obtained results were thoroughly discussed. Only the conclusion lacks a take-home message and simply repeats the findings. Aside from the existing sentences, the conclusion should provide a single clear message followed by a suggestion for future research.
I suggest author to revise their manuscript and re-submit it after minor revisions.
Answer: First, we would like to thank you for the positive and constructive comments and suggestions. All authors have read and approved the manuscript. All necessary corrections have been made, which can be viewed as track changes. We have revised the text format according to the journal guidelines. All "in vivo" and "in vitro" have been changed to italics, and all subheadings (Discussion section) have been changed to capitalization. Conclusion has been modified and provided the meaning of the results and the take-home message.We hope that the corrections will meet with approval.
Round 2
Reviewer 2 Report
I carefully evaluated this revised version of the manuscript, all sections were improved by the authors according to reviewer comments, so I recommend this manuscript for publishing in the Microorganisms Journal in the present form!